# Optimal Quaternary Hermitian LCD Codes

**DOI:** 10.3390/e26050373

**Published:** 2024-04-28

**Authors:** Liangdong Lu, Ruihu Li, Yuezhen Ren

**Affiliations:** Fundamentals Department, Air Force Engineering University, Xi’an 710051, China; liruihu@aliyun.com (R.L.); renyzlw@163.com (Y.R.)

**Keywords:** quaternary code, Hermitian, linear complementary dual, linear code, optimal

## Abstract

Linear complementary dual (LCD) codes, which are a class of linear codes introduced by Massey, have been extensively studied in the literature recently. It has been shown that LCD codes play a role in measures to counter passive and active side-channel analyses on embedded cryptosystems. In this paper, tables are presented of good quaternary Hermitian LCD codes and they are used in the construction of puncturing, shortening and combination codes. The results of this, including three tables of the best-known quaternary Hermitian LCD codes of any length n≤25 with corresponding dimension *k*, are presented. In addition, many of these quaternary Hermitian LCD codes given in this paper are optimal and saturate the lower or upper bound of Grassl’s code table, and some of them are nearly optimal.

## 1. Introduction

Let *q* be a power of a prime *p*, $F_{q}$ be a finite field with *q* elements, and $F_{q}^{n}$ be an *n*-dimensional vector space over $F_{q}$. A *q*-ary ${\left[n,k,d\right]}_{q}$ linear code over $F_{q}$ is a *k*-dimensional subspace of $F_{q}^{n}$ with Hamming distance *d*. For a given ${\left[n,k\right]}_{q}$ linear code, the code $C^{⊥}=\left\{x|x\cdot c=0,c\in C\right\}$ is called the dual code of C. A *q*-ary linear code C is called a linear complementary dual (LCD) code if it meets its dual trivially, that is, $C\cap C^{⊥}=\left\{0\right\}$, which was given by Massey [1,2]. In addition to their applications in data storage, communication systems, and consumer electronics, LCD codes have recently been employed in cryptography and quantum error correcting. Carlet and Guilley in ref. [3] showed that LCD codes play an important role in armoring implementations against side-channel attacks and presented several constructions of LCD codes.

In [4], according to finite geometry theory, Lu et al. proposed the *radical codes* R(C) of C and $C^{⊥}$, which are $R\left(C\right)=C\cap C^{⊥}$. If $R\left(C\right)=C\cap C^{⊥}=\left\{0\right\}$, then C is called a zero radical code, which is the same as the LCD code presented in [2]. Using these zero radical codes, they constructed families of maximal entanglement entanglement-assisted quantum error-correcting codes, which can help to engineer more reliable quantum communication schemes and quantum computers. Furthermore, constructions of Hermitian zero radical BCH codes were discussed in [5], which are also called reversible codes in [1] or LCD cyclic codes in [6]. Güneri et al. studied quasi-cyclic LCD codes and introduced Hermitian LCD codes [7]. Moreover, for the Euclidean case, the question of when cyclic codes are LCD codes is answered affirmatively by Yang and Massey in [8]. Ding et al. investigated LCD cyclic codes in [6], in which several families of LCD cyclic codes were constructed. It is shown that some LCD cyclic codes are optimal linear codes or have the best possible parameters for cyclic codes. Shi et al. constructed a lot of good LCD codes [9,10,11,12]. Moreover, many works have focused on the construction of LCD codes with good parameters, see [13,14,15,16,17,18,19,20,21].

Recently, Carlet, Mesnager, Tang, Qi and Pellikaan in [22] have shown that any [n,k,d]-linear code over $F_{q^{2}}$ is equivalent to an [n,k,d]-linear Hermitian LCD code over $F_{q^{2}}$ for q>2. Araya, Harada and Saito in [23] gave some conditions for the nonexistence of quaternary Hermitian linear complementary dual codes with large minimum weights. Inspired by these works and extending our previous work in [4], we study constructions of linear Hermitian LCD codes over $F_{4}$. Then, some families of linear Hermitian LCD codes with good parameters are constructed from the known optimal codes via puncturing, extending, shortening and the combination method. Compared with the tables of best known linear codes (referred to as the *Database* later) maintained by Markus Grassl in [24], some of our codes presented in this paper saturate the lower bound of Grassl’s code table.

In this paper, an optimal quaternary Hermitian LCD code [18,7,9] is given, which improves the minimal distance of the codes in [4,25,26]. According to classification codes in [27], there exist some optimal quaternary Hermitian LCD codes [15,4,8], [16,4,9], [17,4,10], [19,4,13], [23,5,14] and [15,6,7]. According to [24], the following quaternary Hermitian LCD codes we give in this section are also optimal linear codes: [n,k,d] for 21≤n≤25 and 16≤k≤18; [n−i,15−i,d] for 21≤n≤24 and 0≤i≤2; and [23,4,15], [24,5,15], [21,6,12], [22,6,12], [23,6,13], [24,6,14], and [25,8,12].

This paper is organized as follows. In Section 2, we provide some required basic knowledge on Hermitian LCD codes. We derive constructions of Hermitian LCD codes in Section 3. In Section 4, we discuss Hermitian LCD codes with good parameters.

## 2. Preliminary

In this section, we introduce some basic concepts on quaternary linear codes. Let $F_{4}=\left\{0,1,\omega ,\varpi \right\}$ be the Galois field with four elements, with $\varpi =1+\omega =\omega^{2},\omega^{3}=1$. Denote the *n*-dimensional space over $F_{4}$ by $F_{4}^{n}$; we call a *k*-dimensional subspace C of $F_{4}^{n}$ a *k*-dimensional linear code of length *n* and denote it as C=[n,k]. A matrix *G* whose rows form the basis of C is called a generator matrix of C. If the minimum distance of C is *d*, then C can be denoted as C=[n,k,d]. A code C=[n,k,d] is an *optimal* code if there is no [n,k,d+1] code. An *optimal* code is denoted $\left[n,k,d_{o}\left(n,k\right)\right]$ in this paper. For a given code [n,k,d], if *d* is the largest value present known, then C is called the *best-known code and also denoted as $\left[n,k,d_{o}\left(n,k\right)\right]$.* Denote $d_{l}\left(n,k\right)=max\left\{d\right|$ as an [n,k,d] LCD code}. If a $C=[n,k,d_{l}\left(n,k\right)]$ LCD code saturates the lower or upper bound of Grassl’s code table [24], we call C an optimal LCD code and can say $d_{l}\left(n,k\right)=d_{o}\left(n,k\right)$. If $d_{l}\left(n,k\right)=d_{o}\left(n,k\right)-1$, we call C a nearly optimal LCD code.

Define the Hermitian inner product of *u*, v∈ $F_{4}^{n}$ as$${\left(u,v\right)}_{h}=uv^{2}=u_{1}\bar{v_{1}}+u_{2}\bar{v_{2}}+\cdot \cdot \cdot +u_{n}\bar{v_{n}}.$$

The Hermitian dual code of C=[n,k] is $C^{⊥h}$$=\{x|{\left(x,y\right)}_{h}=0,\forall y\in C\}$, and $C^{⊥h}$=[n,n−k]. A generator matrix $H=H_{(n-k)\times n}$ of $C^{⊥h}$ is called a parity check matrix of C. If C ⊆ $C^{⊥h}$, C is called a *weakly self-orthogonal* code. If C is a self-orthogonal code, then each generator matrix *G* of C must satisfy rank($GG^{†})=0$, where $G^{†}$ is the conjugate transpose of *G*.

If C∩$C^{⊥h}=\left\{0\right\}$, then C (or $C^{⊥h}$) is called a *quaternary Hermitian LCD* code, and each generator matrix *G* of C must satisfy k= rank($GG^{†})$, see refs. [2,4].

In the following sections, we will discuss the construction of Hermitian quaternary LCD code C=[n,k,d], where *d* is as large as possible for a given *n* and k≤5. Firstly, we present some notation for later use.

Let $1_{n}$ = ${\left(1,1,\ldots ,1\right)}_{1\times n}$ and $0_{n}$ = ${\left(0,0,\ldots ,0\right)}_{1\times n}$ denote an all-one vector and an all-zero vector of length *n*, respectively.

Construct$$S_{2}=\left(\begin{array}{ccccc} 0 & 1 & 1 & 1 & 1\\ 1 & 0 & 1 & \omega & \varpi \end{array}\right)=\left(\alpha_{1},\ldots ,\alpha_{5}\right),$$$$S_{3}=\left(\begin{array}{ccccc} S_{2} & 0_{2\times 1} & S_{2} & S_{2} & S_{2}\\ 0_{5} & 1 & 1_{5} & \omega 1_{5} & \varpi 1_{5} \end{array}\right)=\left(\beta_{1},\beta_{2},\cdots ,\beta_{21}\right),$$$$S_{4}=\left(\begin{array}{ccccc} S_{3} & 0_{3\times 1} & S_{3} & S_{3} & S_{3}\\ 0_{21} & 1 & 1_{21} & \omega 1_{21} & \varpi 1_{21} \end{array}\right)=\left(\gamma_{1},\gamma_{2},\cdots ,\gamma_{85}\right).$$$$\begin{array}{c} S_{5}=\left(\begin{array}{ccccc} S_{4} & 0_{4\times 1} & S_{4} & S_{4} & S_{4}\\ 0_{85} & 1 & 1_{85} & \omega 1_{85} & \varpi 1_{85} \end{array}\right)=\left(\zeta_{1},\zeta_{2},\cdots ,\zeta_{341}\right).\\ \vdots \end{array}$$


$$S_{k}=\left(\begin{array}{ccccc} S_{k-1} & 0_{k-1\times 1} & S_{k-1} & S_{k-1} & S_{k-1}\\ 0_{\frac{4^{\left(k-1\right)}}{3}} & 1 & 1_{\frac{4^{\left(k-1\right)}}{3}} & \omega 1_{\frac{4^{\left(k-1\right)}}{3}} & \varpi 1_{\frac{4^{\left(k-1\right)}}{3}} \end{array}\right)$$


It is well known that the matrix $S_{2}$ generates the [5,2,4] simplex code with weight polynomial $1+15y^{4}$. $S_{3}$ generates the [21,3,16] simplex code with weight polynomial $1+63y^{16}$. $S_{4}$ generates the [85,4,64] simplex code with weight polynomial $1+255y^{64}$, $S_{5}$ generates the [341,5,256] simplex code with weight polynomial $1+1023y^{256}$, and $S_{k}S_{k}^{†}=0$ for k=2,3,4,5,⋯, see ref. [28].

### Notation

In the following sections, the conjugation is defined by $\bar{x}=x^{2}$ for $x\in F_{4}$. We use 2 and 3 to represent ω and ϖ in each generator matrix of linear codes, respectively. An ${\left[n,k,d\right]}_{4}$ code is denoted as [n,k,d] for short.

## 3. Hermitian LCD Linear Codes over $F_{4}$


In this subsection, we discuss the construction of [n,k] optimal Hermitian quaternary LCD codes. For k≥5 and n≤20, there are some Hermitian quaternary LCD codes in [21, 23, 26, 28]. For 20≤n≤25, there is no systematic discussion in the literature. The discussion is presented in four cases for n≤25.

**Lemma 1 ([4,29]).** 
*If 21≤n≤25 and 1≤k≤5, then $d_{l}\left(2,24\right)=18$, $d_{l}\left(2,25\right)=19$, $d_{l}\left(3,21\right)=15$, $d_{l}\left(3,22\right)=15$, $d_{l}\left(4,22\right)=14$, $d_{l}\left(4,23\right)=15$, $d_{l}\left(5,24\right)=15$. All the other Hermitian quaternary LCD codes saturate the lower bound of Grassl’s code table [24].*


**Proof.** Refs. [4,29] proved this lemma. □

**Lemma 2 ([30]).** 
*There exist [n,n−2,2] and [n,n−3,2] quaternary Hermitian LCD codes.*


**Proof.** (1) For when *n* is even, let $G=\left[I_{2}|\frac{1_{1\times n}}{0_{1\times n}}\right]$. If *G* is a check matrix of *C* with generator matrix *H*, then C=[n,n−2,2] and $rank(HH^{h})=n-2$. For when *n* is even, let $G=\left[I_{2}\left(\begin{array}{c} \varpi \\ \varpi \end{array}\right)|\frac{1_{1\times n}}{1\times n}\right]$. If *G* is a check matrix of *C* with generator matrix *H*, then C=[n,n−2,2] and $rank(HH^{h})=n-2$.(2) For when *n* is odd, let $G=\left[I_{3}|\frac{1_{1\times n}}{0_{2\times n}}\right]$. If *G* is a check matrix of *C* with generator matrix *H*, then C=[n,n−3,2] and $rank(HH^{h})=n-3$. For when *n* is even, let $G=\left[I_{3}\left(\begin{array}{c} \varpi \\ \varpi \\ 0 \end{array}\right)|\frac{1_{1\times n}}{0_{2\times n}}\right]$. If *G* is a check matrix of *C* with generator matrix *H*, then C=[n,n−3,2] and $rank(HH^{h})=n-3$. □

**Theorem 1.** 
*If 21≤n≤25 and 13≤k≤18, then there exist 29 optimal quaternary Hermitian LCD codes saturating the lower bound of Grassl’s code table [24], as in Table 1.*


**Proof.** For 21≤n≤25, calculating by Magma, one can obtain nine optimal LCD codes as follows: [21,14,5], [21,15,5], [21,16,4], [22,16,4], [24,16,6], [22,18,3], [23,18,4], [25,18,5], [23,19,3].And then, calculating by Magma, one can obtain another five optimal codes, [27,19,6], [26,21,4], [25,18,5], [26,20,4], [33,24,6], which are not all quaternary Hermitian LCD codes.Case 1. Construction of quaternary Hermitian LCD codes via puncturing. Puncturing C=[23,15,6] on coordinate sets {16}, {1,19}, one can obtain [22,15,5] and [21,17,3] Hermitian quaternary LCD codes. Puncturing C=[24,17,5] on coordinate sets {1,18}, one can obtain the [22,17,4] Hermitian quaternary LCD code. Puncturing C=[24,19,4] on coordinate sets {1,4,8}, one can obtain the [21,19,2] quaternary Hermitian LCD code.Case 2. Construction of LCD codes via shortening. Shortening C=[27,19,6] on coordinate sets {1,4}, {1,2,3,8}, {1,2,3,4,7}, {1,2,3,4,7,8}, one can obtain the [25,17,6], [23,15,6], [22,14,6], [21,13,6] Hermitian quaternary LCD codes, respectively. Shortening D=[26,21,4] on coordinate sets {1}, {1,2} obtain [25,20,4] and [24,19,4] Hermitian quaternary LCD codes, respectively. Shortening D=[25,18,5] on coordinate sets {4} and {1,4}, one can obtain the [24,17,5], [23,16,5] Hermitian quaternary LCD codes. Shortening D=[26,20,4] on coordinate sets {1}, {1,2}, {1,2,4}, one can obtain the [25,19,4], [24,18,4], [23,17,4] Hermitian quaternary LCD codes. Shortening D=[33,24,6] on coordinate sets {1,2,3,4,5,6,7,8} and {1,2,3,4,5,6,7,8,9}, one can obtain the [25,16,6] and [24,15,6] Hermitian quaternary LCD codes. Shortening D=[33,24,6] on coordinate sets {1,2,3,4,5,6,7,8,9,10} and {1,2,3,4,5,6,7,8,9,10,14}, one can obtain the C=[23,14,6] and [22,13,6] Hermitian quaternary LCD codes. □

**Remark 1.** 
*In Theorem 2, all of the codes are optimal quaternary Hermitian LCD codes. Since [21,3,16] is a simplex code, there is no [21,18,3] quaternary Hermitian LCD code. Hence, [21,18,2] is an optimal quaternary Hermitian LCD code. By shortening D=[33,24,6] on coordinate sets {1,2,3,4,5,6,7,8,9,10,11,12}, we can obtain C=[21,12,6]. This is a nearly optimal quaternary Hermitian LCD code with weight enumerator $1+279z^{6}+1116z^{7}+5739z^{8}+22023z^{9}+79815z^{10}+\ldots$.*


**Theorem 2.** 
*If 21≤n≤25 and 6≤k≤8, then $d_{l}\left(6,21\right)=12$, $d_{l}\left(6,22\right)=12$, $d_{l}\left(6,23\right)=13$, $d_{l}\left(6,24\right)=14$, $d_{l}\left(8,25\right)=12$, $d_{l}\left(7,20\right)=10$. All these codes are quaternary Hermitian LCD codes saturating the lower or upper bound of Grassl’s code table.*


**Proof.** A constacyclic code C=[21,15,5] is given in [21], where its generator polynomial is $x^{6}+\bar{\omega }x^{5}+x^{4}+\bar{\omega }x^{2}+x+\bar{\omega }$. The dual code of C is the code D=[21,6,12] with a generator matrix $G_{6,21}$, and both C and D are quaternary Hermitian LCD codes.Let$G_{6,21}=\left(\begin{array}{c} 211210221102122100000\\ 303201310313021010000\\ 221130310133220001000\\ 022113031013322000100\\ 320131031302101000010\\ 223203322032332000001 \end{array}\right)$,$G_{6,24}=\left(\begin{array}{c} 100000111120112310122020\\ 010000011112011231012202\\ 001000201111101123201220\\ 000100120111310112020122\\ 000010112011231011202012\\ 000001111201123101220201 \end{array}\right)$,$G_{7,20}=\left(\begin{array}{c} 11011111101000000011\\ 11120323003110000012\\ 30132223011030000020\\ 11202023001001100001\\ 11000132002013010001\\ 30220233000013001003\\ 30313002003012000101 \end{array}\right)$,$G_{8,25}=\left(\begin{array}{c} 1000000010012123213103310\\ 0100000031322111123203311\\ 0010000012120332303223021\\ 0001000021122001100022303\\ 0000100013332221330202233\\ 0000010010113203310220220\\ 0000001001011320330022022\\ 0000000112200212232000033 \end{array}\right)$,There exists a quaternary Hermitian LCD code [24,6,14] with generator matrix $G_{6,24}$. Its weight enumerator is $1+207z^{14}+378z^{15}+630z^{16}+360z^{17}+495z^{18}+1062z^{19}+585z^{20}+180z^{21}+162z^{22}+36z^{23}$. Puncturing C=[24,6,14] on coordinate sets {7}, {1,3}, we can obtain two quaternary Hermitian LCD codes: [23,6,13], [22,6,12].There exists a quaternary Hermitian LCD code [20,7,10] with generator matrix $G_{7,20}$. Its weight enumerator is $1+210z^{10}+594z^{11}+969z^{12}+1647z^{13}+2703z^{14}+3519z^{15}+3060z^{16}+2205z^{17}+1107z^{18}+291z^{19}+78z^{20}$.There exists a quaternary Hermitian LCD code [25,8,12] with generator matrix $G_{8,25}$. Its weight enumerator is $1+177z^{12}+540z^{13}+1365z^{14}+2721z^{15}+4836z^{16}+8283z^{17}+10938z^{18}+11694z^{19}+10983z^{20}+7734z^{21}+4185z^{22}+1617z^{23}+411z^{24}+25z^{25}$.Shortening the [25,8,12] quaternary Hermitian LCD code on coordinate sets {2}, one can obtain [24,7,12]. Its weight enumerator is $1+102z^{12}+267z^{13}+561z^{14}+1086z^{15}+1764z^{16}+2628z^{17}+3144z^{18}+2730z^{19}+2226z^{20}+1233z^{21}+495z^{22}+120z^{23}+27z^{24}$. We can deduce a submatrix $G_{7,25}$ from $G_{8,25}$. Setting $G_{7,25}$ as a generator matrix, one can obtain [25,7,12]. □

**Theorem 3.** 
*If 21≤n≤25 and 8≤k≤15, then there exist 27 quaternary Hermitian LCD codes, as in Table 2.*


**Proof.** Let$A_{12}^{⊤}=\left(\begin{array}{c} 300201200120300312\\ 330322220222130112\\ 133330122212313132\\ 213031312011331230\\ 002033213230313123\\ 021303131201133123\\ 300300221033231123\\ 330133122213123323\\ 033110212131112103\\ 203212121103311021\\ 020120012030031210\\ 002012001203003121 \end{array}\right)$,$A_{11}^{⊤}=\left(\begin{array}{c} 1232130030332100\\ 3202331131102132\\ 1100132302320200\\ 1321332231223320\\ 0013213203303002\\ 0131021110330323\\ 1322322221212333\\ 2203012112300132\\ 3122131011102120\\ 0002212120320222\\ 3223331032121221 \end{array}\right)$.There exists a code [30,18,8] with generator matrix $G_{18,30}=\left[I_{18}|A_{12}\right]$. It is not a quaternary Hermitian LCD code. Shortening C=[30,18,8] on coordinate sets {1,3,6,12,13,17}, {1,2,3,4,5,6,8}, {1,2,3,4,5,6,7,8} and {1,2,3,4,5,6,7,8,10}, one can obtain quaternary Hermitian LCD codes [24,12,8], [23,11,8], [22,10,8] and [21,9,8], respectively.There exists a code [27,16,7] with generator matrix $G_{16,27}=\left[I_{16}|A_{11}\right]$. Shortening C=[27,16,7] on coordinate sets {1,2}, {1,2,3}, {1,2,3,4}, {1,2,3,4,5} and {1,2,3,4,5,6}, one can obtain quaternary Hermitian LCD codes [25,14,7], [24,13,7], [23,12,7], [22,11,7] and [21,10,7], respectively.Let $A_{10}^{⊤}=\left(\begin{array}{c} 11121122120230323210\\ 00000000000000000000\\ 12103313201003102111\\ 11331313032000112032\\ 21133320003300003220\\ 01030322203233222121\\ 32210120113300112032\\ 00331212201211230320\\ 11321302013232210003\\ 11010110212023020323 \end{array}\right)$,$B_{16}=\left(\begin{array}{c} 0331200330200\\ 0033102033020\\ 2330330112100\\ 0233003311210\\ 1201303302020\\ 2213122121000\\ 0221321212100\\ 0022113221210\\ 1220202211020\\ 2211033130300\\ 0221130313030\\ 3133123113000\\ 0313331211300\\ 0031313321130\\ 3112111310210\\ 1133222202120 \end{array}\right)$,There exists a code [30,20,6] with generator matrix $G_{20,30}=\left[I_{20}|A_{10}\right]$. Shortening C=[30,20,6] on coordinate sets {1,2,3,4,5}, {1,2,3,4,5,6}, {1,2,3,5,6,7,13}, {1,2,3,4,5,6,7,8,9} and {1,2,3,4,5,6,7,8}, one can obtain quaternary Hermitian LCD codes [25,15,6], [24,14,6], [23,13,6], [22,12,6] and [21,11,6], respectively.There exists a code [29,16,8] with generator matrix $G_{16,29}=\left[I_{16}|B_{16}\right]$. It is not a quaternary Hermitian LCD code. Shortening C=[29,16,8] on coordinate sets {1,2,3,4}, {1,2,3,4,6}, {1,2,3,4,5,6}, {1,2,3,4,5,6,8}, {1,2,3,4,5,6,7,8}, one can obtain quaternary Hermitian LCD codes [25,12,8], [24,11,8], [23,10,8], [22,9,8] and [21,8,8], respectively.Let $A_{14}=\left(\begin{array}{c} 13000123002211\\ 10211103211330\\ 01021110321133\\ 33231222301220\\ 03323122230122\\ 22110130001230\\ 02211013000123\\ 33112232033321\\ 12200332312223\\ 32113300102111\\ 12300221101300\\ 01230022110130\\ 00123002211013\\ 33321033112232\\ 21110321133001\\ 11111111111111 \end{array}\right)$,$A_{15}=\left(\begin{array}{c} 102333010301203\\ 210033301030123\\ 122000202121223\\ 013221122033211\\ 202131213310100\\ 220013121331010\\ 223330023223223\\ 123212100103011\\ 113310121300022\\ 212032120112232\\ 322220031221030\\ 033023210111210\\ 000220311212022\\ 000312113202231\\ 000121022211230\\ 000102331310131 \end{array}\right)$,$A_{12}=\left(\begin{array}{c} 333231302012\\ 122021230233\\ 012231331333\\ 110121333101\\ 322213123331\\ 123133323323\\ 321131111021\\ 301331322211\\ 212033023133\\ 021232013200\\ 113202212020\\ 000012102221\\ 000001110222 \end{array}\right)$,There exists a code [30,16,9] with generator matrix $G_{16,30}=\left[I_{16}|A_{14}\right]$. It is not a quaternary Hermitian LCD code. Shortening C=[30,16,9] on coordinate sets {1,2,3,4,5}, {1,2,3,4,5,11}, {1,2,3,4,5,6,11} and {1,2,3,4,5,6,11}, one can obtain quaternary Hermitian LCD codes [25,11,9], [24,10,9], [23,9,9] and [22,8,9], respectively.There exists a code [31,16,10] with generator matrix $G_{16,31}=\left[I_{16}|A_{15}\right]$. It is not a quaternary Hermitian LCD code. Shortening C=[31,16,10] on coordinate sets {8,9,10,11,16}, {1,6,9,10,12,15}, {2,4,5,9,10,13,15} and {3,4,7,8,11,12,13,15}, one can obtain quaternary Hermitian LCD codes [26,11,10], [25,10,10], [24,9,10] and [23,8,11], respectively. Puncturing C=[23,8,11] on coordinate sets {2} and {1,2}, one can obtain quaternary Hermitian LCD codes [22,8,10] and [21,8,9], respectively.There exists a quaternary Hermitian LCD code [25,13,7] with generator matrix $G_{13,25}=\left[I_{13}|A_{12}\right]$. □

**Theorem 4.** 
*$d_{l}\left(7,21\right)=10$, $d_{l}\left(7,22\right)=11$, $d_{l}\left(7,22\right)=11$, $d_{l}\left(7,23\right)=12$, $d_{l}\left(7,25\right)=13$, $d_{l}\left(6,25\right)=14$, $d_{l}\left(12,25\right)=8$, $d_{l}\left(13,25\right)=7$, $d_{l}\left(7,18\right)=9$ and $d_{l}\left(7,19\right)=9$ are Hermitian quaternary LCD codes.*


**Proof.** Let$G_{7,24}=\left(\begin{array}{c} 022312110003211100000321\\ 102231211000321110000032\\ 110223121100032121000003\\ 211022311110003232100000\\ 121102232111000303210000\\ 312110223211100000321000\\ 231211020321110000032100 \end{array}\right)$,$G_{7,25}=\left(\begin{array}{c} 1100000032031300131303121\\ 0010000003203130213130311\\ 0001000003123223332223321\\ 0000100033111212020312021\\ 0000010020112011011101111\\ 0000001011212331312020201\\ 0000000112322303222332331 \end{array}\right)$,$G_{7,19}=\left(\begin{array}{c} 1130121231010000000\\ 0133203320031200000\\ 0303332320310100000\\ 0131030320010011000\\ 0130001110020230300\\ 0301202120000230010\\ 0302110010030220003 \end{array}\right)$,

$G_{6,25}=\left(\begin{array}{c} 3111201330100001230133122\\ 1111110133030000122013312\\ 2311100013323000012201331\\ 1131131001312300001220133\\ 1213130100101230003122013\\ 1121313010000123003312201 \end{array}\right)$

There exists a code [24,7,13] with generator matrix $G_{7,24}$. Its weight enumerator is $1+384z^{13}+744z^{14}+888z^{15}+1746z^{16}+2544z^{17}+3156z^{18}+2928z^{19}+2118z^{20}+1200z^{21}+540z^{22}+120z^{23}+15z^{24}$. It is not a quaternary Hermitian LCD code. Puncturing $C_{1}$ on coordinate sets {1}, {1,3}, {1,2,7}, we can obtain [23,7,12], [22,7,11], [21,7,10] quaternary Hermitian LCD codes.There exists a quaternary Hermitian LCD code [25,7,13] with generator matrix $G_{7,25}$. Its weight enumerator is $1+189z^{13}+495z^{14}+750z^{15}+1179z^{16}+1908z^{17}+2577z^{18}+2967z^{19}+2667z^{20}+1932z^{21}+1092z^{22}+495z^{23}+117z^{24}+15z^{25}$.There exists a quaternary Hermitian LCD code [25,6,14] with generator matrix $G_{6,25}$. Its weight enumerator is $1+48z^{14}+240z^{15}+432z^{16}+534z^{17}+573z^{18}+648z^{19}+657z^{20}+510z^{21}+363z^{22}+84z^{23}+6z^{24}$.There exists an optimal quaternary Hermitian LCD code [19,7,9] with generator matrix $G_{7,19}$. Its weight enumerator is $1+195z^{9}+483z^{10}+888z^{11}+1479z^{12}+2361z^{13}+3165z^{14}+3327z^{15}+2508z^{16}+1368z^{17}+492z^{18}+117z^{19}$. Puncturing the quaternary Hermitian LCD code [19,7,9] on coordinate sets {1}, one can obtain the optimal quaternary Hermitian LCD code [18,7,9] with weight enumerator $1+393z^{9}+666z^{10}+1245z^{11}+2193z^{12}+3315z^{13}+3597z^{14}+2799z^{15}+1554z^{16}+504z^{17}+117z^{18}$. □

## 4. Discussion and Conclusions

This paper is dedicated to the construction of quaternary Hermitian LCD codes. For k≤n and n≤25, each [n,k] quaternary Hermitian LCD code is constructed. Some of these quaternary Hermitian LCD codes constructed in this paper are optimal codes which saturate the bound of the minimum distance of the code table in [24], and some of them are nearly optimal codes. According to weight enumerators for classification codes in [27], there exist some optimal codes, [15,4,9], [16,4,10], [17,4,11], [19,4,14], and [23,5,15], which are not LCD codes. In addition, the number of these five optimal codes is one. Thus, the [15,4,8], [16,4,9], [17,4,10], [19,4,13], and [23,5,14] quaternary Hermitian LCD codes in this paper are optimal. In [27], all of the codes with parameters of [15,6,8] are self-orthogonal. Thus, the quaternary Hermitian LCD code in this paper, [15,6,7], is also optimal. We emphasize that there are three quaternary Hermitian LCD codes, [18,7,9], [19,7,9] and [20,7,10], which are optimal.

According to ref. [24], the following quaternary Hermitian LCD codes constructed in this paper are also optimal codes with parameters of [n,k,d] for 21≤n≤25 and 16≤k≤18; [n−i,15−i,d] for 21≤n≤24 and 0≤i≤2: [23,4,15], [24,5,15], [21,6,12], [22,6,12], [23,6,13], [24,6,14], [25,8,12] and [20,7,10]. Except for these codes mentioned above, the quaternary Hermitian LCD codes constructed in this paper do not reach the known upper or lower bounds of the minimum distance of a linear code. Nonetheless, the minimum distances of these codes appears to be the best possible. These codes are the best possible among those obtainable by our approach.

Combining the results in the previous subsections, we improved the table of lower and upper bounds on the minimum distance of quaternary Hermitian LCD codes for n≤20 [4,25,30] in Table 3. In addition, many lower and upper bounds of the minimal distance of Hermitian LCD codes with a length of n≤25 are listed. To make the bounds in Table 3 tighter, we need to choose other quaternary Hermitian LCD codes better than those given in this paper and investigate other code constructions to raise the lower bounds. We also plan to explore the construction of Hermitian LCD codes from a geometric aspect to decrease the upper bounds.

In [4,20,25], it has been shown that if there exists a quaternary Hermitian [n,k,d] code over $F_{q^{2}}$, then there exists a maximal entanglement entanglement-assisted quantum error correcting code (EAQECC) over $F_{q}$ with parameters [[n,2k−n+c,d;c]], where *c* is the rank of the product of the parity check matrix and its conjugate. Moreover, a maximal entanglement EAQECC derived from an LCD code has the same minimum distance as the underlying classical code. Hence, all of the optimal quaternary Hermitian LCD codes can be used to construct optimal binary maximal entanglement EAQECCs. In addition, from the three quaternary Hermitian LCD codes [18,7,9] given in this paper, a maximal entanglement EAQECC [[18,7,9;11]] can be constructed, which improves the minimal distance of the codes in [4,25]. The maximal entanglement EAQECCs [[19,7,9;12]] and [[20,7,10;12]] are optimal and are different to the codes constructed in [30].

## Figures and Tables

**Table 1 entropy-26-00373-t001:** **Optimal quaternary Hermitian LCD codes with 21≤n≤25 and 13≤k≤19.**

n∖k	13	14	15	16	17	18	19
21	6	5	5	4	3	2	2
22	6	6	5	4	4	3	2
23		6	6	5	4	4	3
24			6	6	5	4	4
25				6	6	5	4

**Table 2 entropy-26-00373-t002:** **Optimal quaternary Hermitian LCD codes with 21≤n≤25 and 8≤k≤15.**

n∖k	8	9	10	11	12	13	14	15
21	9	8	7	6				
22	10	8	8	7	6			
23	11	9	8	8	7	6		
24	11	10	9	8	8	7	6	
25		11	10	9	8	7	7	6

**Table 3 entropy-26-00373-t003:** **Lower and upper bounds on the minimum distance of quaternary Hermitian LCD codes. The bold entries represent improvements over prior work. The superscript * represents the codes that achieve bounds given in the Grassl table.**

n∖k	1	2	3	4	5	6	7	8	9	10	11	12
3	3 *	2 *										
4	3 *	2	1									
5	5 *	3	2	2 *								
6	5 *	4 *	3	2 *	1							
7	7 *	5 *	4 *	3 *	2 *	2 *						
8	7 *	6 *	5 *	4 *	3 *	2 *	1					
9	9 *	6	6 *	5 *	4 *	3 *	2 *	2 *				
10	9 *	7	6 *	6 *	5 *	4 *	3 *	2 *	1	1		
11	11 *	8 *	7 *	6 *	6 *	5 *	4 *	3 *	2 *	2 *	1	
12	11 *	9 *	8 *	7 *	6 *	5	4 *	4 *	3 *	2 *	2 *	1
13	13 *	10 *	9 *	8 *	7 *	6 *	5 *	4 *	4 *	3 *	2 *	2 *
14	13 *	10	9	8	7–8	7 *	6 *	5 *	4 *	4 *	3 *	2 *
15	15 *	11	10	9	8 *	7	7 *	6 *	5 *	4 *	4 *	3 *
16	15 *	12 *	11	10	9 *	8 *	7–8	6–7	6 *	5 *	4 *	4 *
17	17 *	13 *	12 *	11	10 *	9 *	7–8	7–8	6–7	6 *	5 *	4 *
18	17 *	14 *	13 *	11–12	10 *	9–10	**9 ***	8 *	7–8	6 *	5–6	5 *
19	19 *	14	13	12 *–13	11 *	10 *	9 *	8 *–9	8 *	7 *	6 *	5–6
20	19 *	15	14	13 *	12 *	11 *	10 *	9 *	8 *–9	7–8	6–7	6 *
21	21 *	16 *	15	14 *	12	12 *	10–11	9–10	8–9	7–9	6–8	6–7
22	22 *	17 *	15	14	13	12 *–13	11–12	**10**	8–10	8–9	7–9	6–8
23	23 *	18 *	16 *	15	14	13 *	12 *–13	11	9–11	8–10	8–9	7–9
24	24 *	18	17 *	16 *	15	14 *	12–13	11–13	10–12	9–11	8–10	8–9
25	25 *	19	18 *	17 *	15	14–15	13–14	12 *–13	11–13	10–12	9–11	8–10
n∖k	13	14	15	16	17	18	19	20	21	22	23	24
14	1											
15	2 *	2 *										
16	3 *	2 *	1									
17	3–4	3 *	2 *	2 *								
18	4 *	3 *	3 *	2 *	1							
19	5 *	4 *	3 *	3 *	2 *	2 *						
20	5–6	5 *	4 *	3 *	2	2 *	1	1				
21	6 *	5 *–6	5 *	4 *	3 *	2	2 *	2 *	1			
22	6 *–7	6 *	5 *–6	4 *–5	4 *	3 *	2 *	2 *	2	1		
23	6–8	6 *–7	6 *	5 *–6	4 *–5	4 *	2 *	2 *	2 *	2 *	1 *	
24	7–9	6–8	6 *–7	6 *	5 *–6	4 *–5	3 *	3 *	2 *	2 *	2 *	1 *
25	7–9	7–9	6–8	6 *–7	6 *	5 *–6	4 *	4 *	3 *	2 *	2 *	2 *

## Data Availability

The authors confirm that the data supporting the findings of this study are available within the manuscript.

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
