# Peer review of "Optimal Quaternary Hermitian LCD Codes"

_entropy, 2024, doi:10.3390/e26050373_

Round 1
Reviewer 1 Report
Comments and Suggestions for Authors
Title: Optimal Quaternary Hermitian LCD codes
In this manuscript, the authors study quaternary Hermitian LCD codes. The research approach is trivial in my opinion. From already obtained codes located in databases and easily accessible with puncturing and shortening, the authors construct other LCD codes. For the calculations, they use the MAGMA software package and not their own methods and algorithms.
The obtained results are also not of particular interest because in most cases there are already known codes with such parameters. There is no theoretical result either.
For these reasons, I cannot recommend this article for publication in the MDPI journal Entropy.
Comments on the Quality of English LanguageNo comments
Author Response
Dear Prof. We would like to thank you for your valuable comments that have been helpful to improve the paper considerably. We hope to study new quaternary Hermitian LCD codes using new tools in the future. I look forward to having good academic interaction and exchange with you in the future.Reviewer 2 Report
Comments and Suggestions for Authors
Nowadays Hermitian LCD codes are rather important since they found applications in
cryptology and quantum error correcting coding.
In this paper, the authors systemized known constructions and presented several newly found codes.
Search results are summarized in a complete table of codes of length up to 25.
Remarks:
1. In Table 3, I recommend marking those codes that achieve bounds given in the Grassl table.
2. Typo: Table 3 has the wrong number (Table 2).
I recommend acceptance after minor corrections.
Author Response
We would like to thank you for your kind advises.
We mark those codes that achieve bounds given in the Grassl table,
and revise the number of Table 3.

Reviewer 3 Report
Comments and Suggestions for Authors
I did not go through the proofs. However, it seems that all the results are new and interesting.
Comments on the Quality of English LanguageI suggest the authors to check the language expression one more time.
Author Response
We would like to thank you for your valuable suggestions and check the language expression carefully.
Reviewer 4 Report
Comments and Suggestions for Authors
In this manuscript, the authors aim to find good quaternary Hermitian LCD codes.
There are two things to do: to give quaternary Hermitian LCD codes
and verify they are optimal. The strategy is clear.
The authors present the generator matrices of some LCD codes .
Then, the idea is in the construction of puncturing, shortening and
combination the given codes to obtain more good quaternary Hermitian LCD
codes with different lengths. Results including the best-known quaternary
Hermitian LCD codes of any length $ n \leq 25$ with corresponding
dimension $k$ are presented.
{\bf Conclusions about the paper}
The results are interesting and correct.
In my opinion, this paper
can be accepted after considering some minor revisions.
Some specific comments are given
in the following lines.
1. Page 1, line -10 : " Carlet and Guilley in$^{3}$
showed that LCD codes play an important role" $\rightarrow$
"Carlet and Guilley in Ref.$[3]$
showed that LCD codes play an important role".
Modify in other similar situations
2. Page 2, line -2 : "If there exists an $[n,k,d]$ and $d$ is the largest value present known, then
$\mathcal{C}$ is called a {\it best known } code and also
denoted $[n,k,d_{o}(n,k)]$." $\rightarrow$ "For a given code $[n,k,d]$,
if $d$ is the largest value present known, then
$\mathcal{C}$ is called a {\it best-known } code and also
denoted $[n,k,d_{o}(n,k)]$.".
3. Page 2, line -1 : "If an $\mathcal{C}=[n, k, d_{l}(n,k)]$ LCD
code is..." $\rightarrow$
"If a code $\mathcal{C}=[n, k, d_{l}(n,k)]$ LCD
code is...".
4. Page 3, line 5: "$({ u,v})={ uv^{h}}=u_{1}\bar{v_{1}}+u_{2}\bar
{v_{2}}+\cdot\cdot\cdot+u_{n}\bar{v_{n}}.$" and "$({ u,v})_{h}={ uv^{2}}=u_{1}\bar{v_{1}}+u_{2}\bar
{v_{2}}+\cdot\cdot\cdot+u_{n}\bar{v_{n}}.$".
5. Page 3, line 8: "If
$C$ is a self-orthogonal code then each generator matrix..." $\rightarrow$
"If
$C$ is a self-orthogonal code, then each generator matrix...".
6. Page 4, Paragraph 5: "$\mathbf{F}_{4}$"
$\rightarrow$ "$\mathbb{F}_{4}$" . Same modifies in other similar situations
7. Page 12, Paragraph 10:
"According to,22 the following quaternary Hermitian LCD codes constucted in
this paper are also optimal..."
$\rightarrow$ "According to Ref.[22], the following quaternary Hermitian LCD codes constucted in
this paper are also optimal...".
8. Page 12, Paragraph 18: "the minimum distances of those codes appear very
good in general." make me confuse.
$\rightarrow$ "the minimum distances of those codes appear to be the best possible ones ".

The language is very fluent and professional.
Author Response
Reply to Reviewer 4We would like to thank you for your valuable comments that have been helpful to improve the paper considerably. All changes in the manuscript relevant to the comments received are highlighted in Blue.
1. {\bf Comment}: Page 1, line -10 : " Carlet and Guilley in$^{3}$ showed that LCD codes play an important role" $\rightarrow$ "Carlet and Guilley in Ref.$[3]$ showed that LCD codes play an important role". Modify in other similar situations
{\bf Response}:Thanks for your valuable suggestions. We revised as : "Carlet and Guilley in Ref.$[3]$ showed that LCD codes play an important role.". 2. {\bf Comment}: Page 2, line -2 : "If there exists an $[n,k,d]$ and $d$ is the largest value present known, then $\mathcal{C}$ is called a {\it best known } code and also denoted $[n,k,d_{o}(n,k)]$." $\rightarrow$ "For a given code $[n,k,d]$, if $d$ is the largest value present known, then $\mathcal{C}$ is called a {\it best-known } code and also denoted $[n,k,d_{o}(n,k)]$.".
{\bf Response}: We revised as "For a given code $[n,k,d]$, if $d$ is the largest value present known, then $\mathcal{C}$ is called a {\it best-known } code and also denoted $[n,k,d_{o}(n,k)]$." 3. {\bf Comment}: Page 2, line -1 : "If an $\mathcal{C}=[n, k, d_{l}(n,k)]$ LCD code is" $\rightarrow$ "If a code $\mathcal{C}=[n, k, d_{l}(n,k)]$ LCD code is". {\bf Response}: We revised as "If a code $\mathcal{C}=[n, k, d_{l}(n,k)]$ LCD code is'' 4. {\bf Comment}: Page 3, line 5: "$({ u,v})={ uv^{h}}=u_{1}\bar{v_{1}}+u_{2}\bar {v_{2}}+\cdot\cdot\cdot+u_{n}\bar{v_{n}}.$" $\rightarrow$ "$({ u,v})_{h}={ uv^{2}}=u_{1}\bar{v_{1}}+u_{2}\bar {v_{2}}+\cdot\cdot\cdot+u_{n}\bar{v_{n}}.$". {\bf Response}: We have revised "$({ u,v})_{h}={ uv^{2}}=u_{1}\bar{v_{1}}+u_{2}\bar {v_{2}}+\cdot\cdot\cdot+u_{n}\bar{v_{n}}.$". 5. {\bf Comment}: Page 3, line 8: "If $C$ is a self-orthogonal code then each generator matrix..." $\rightarrow$ "If $C$ is a self-orthogonal code, then each generator matrix...".
{\bf Response}: We have revised "If $C$ is a self-orthogonal code, then each generator matrix...".
6. {\bf Comment}: Page 4, Paragraph 5: "$\mathbf{F}_{4}$" $\rightarrow$ "$\mathbb{F}_{4}$" . Same modifies in other similar situations.
{\bf Response}: We have revised. 7. {\bf Comment}: Page 12, Paragraph 10: "According to,22 the following quaternary Hermitian LCD codes constucted in this paper are also optimal..." $\rightarrow$ "According to Ref.[22], the following quaternary Hermitian LCD codes constucted in this paper are also optimal...".
{\bf Response}: We have revised "According to Ref.[22], the following quaternary Hermitian LCD codes constucted in this paper are also optimal". 8. {\bf Comment}: Page 12, Paragraph 18: "the minimum distances of those codes appear very good in general." make me confuse. $\rightarrow$ "the minimum distances of those codes appear to be the best possible ones ". {\bf Response}: We have revised as "the minimum distances of those codes appear to be the best possible ones ".
